# Magnetic Resonance Imaging (MRI) of Spanish Sheep Cheese: A Study on the Relationships between Ripening Times, Geographical Origins, Textural Parameters, and MRI Parameters

**DOI:** 10.3390/foods13203225

**Published:** 2024-10-10

**Authors:** José Segura, María Encarnación Fernández-Valle, Karen Paola Cruz-Díaz, María Dolores Romero-de-Ávila, David Castejón, Víctor Remiro, María Isabel Cambero

**Affiliations:** 1Food Technology Department, Faculty of Veterinary Medicine, Complutense University of Madrid, 28040 Madrid, Spainmdavilah@ucm.es (M.D.R.-d.-Á.); vremiro@ucm.es (V.R.); icambero@ucm.es (M.I.C.); 2ICTS Complutense Bioimaging (BioImac), Research Assistance Centre, Complutense University of Madrid, 28040 Madrid, Spain; evalle@ucm.es (M.E.F.-V.); dcastejon@ucm.es (D.C.)

**Keywords:** sheep cheese, ripening time, manufacturing process, geographical origin, MRI, textural parameter

## Abstract

The evolution of structural changes and the textural features during the ripening process of four varieties of Spanish sheep cheese were studied using Magnetic Resonance Imaging (MRI). Specifically, longitudinal (*T*_1_) and transverse (*T*_2_) relaxation times and apparent diffusion coefficient maps were analyzed. Also, proton density was used to improve the description of the structure of the cheeses. The MRI results displayed important information about cheese matrix structure, associated with different manufacturing processes (industrial vs. traditional), ripening times (RTs, from 2 to 180 days), and geographical origins. A significant interaction between RT and cheese variety related to the variations in physicochemical and textural parameters was found. Linear regression models were developed per the abundant literature. Logarithmic regression models showed the highest R^2^ when monitoring the dependency on *T*_1_ and *T*_2_ parameters of water content, water activity, RT, and some texture parameters. Therefore, these results support that MRI is a useful technology to monitor the ripening process, predict textural behavior and physicochemical variables, and characterize the structure of different varieties of sheep cheese.

## 1. Introduction

A variety of ewe milk cheeses are produced in the Mediterranean area of Europe, including hard and semi-hard cheeses using animal rennet as a coagulant, and they are widely accepted because of their highly desirable sensory qualities. These characteristics include aromas and flavors developed during the ripening process from milk fat, proteins, and carbohydrates [1]. For instance, several cheese varieties are manufactured in Spain.

More specifically, in the north-western (Castilla y León, CL) and central (Castilla-La Mancha, CLM) regions of Spain, several types of semi-hard sheep cheese are made with similar external appearance and close manufacturing technology. Typically, the manufacturing of such varieties uses enzymatic coagulation and, in most cases, the addition of mesophilic starter cultures (generally, strains of *Lactococcus lactis* ssp. *Lactis* and *Lactococcus lactis* ssp. *Cremoris*). Following this, the curd undergoes a compression process and, eventually, an aging period [2].

In both the CL and CLM regions, the cheese is either traditionally (artisanal) or industrially produced. Traditional (T) cheese is manufactured from raw milk in local cheeseries. In contrast, industrial (I) cheese is made either with raw or pasteurized milk at a factory level, thus implying strictly controlled manufacturing conditions. In general, bovine rennet is used for ewe’s milk cheese (EMC) in the CLM region, whereas ovine rennet is used in the CL one (Appendix A).

Manchego, with Protected Designation of Origin (PDO) [3], one of the most popular and exported types of Spanish cheese, is manufactured in the CLM region exclusively from Manchega breed ewe milk. On the contrary, the most known EMC from the CL region is Castellano cheese, which is made from ewe milk of Churra and Castellana breeds [4,5,6].

Desirable textural properties are crucial for consumer acceptance and contribute to the characterization and identification of the type of cheese. Traditionally, cheese texture has been evaluated by sensory and destructive instrumental methods [7].

Magnetic Resonance Imaging (MRI) is a non-invasive and non-destructive technique that provides structural information on biological matrices [8]. The use of MRI generates images of the macroscopic structure. It also allows the quantification of magnetic resonance parameters, such as spin–lattice (*T*_1_) and spin–spin (*T*_2_) relaxation times, and the apparent diffusion coefficient (*ADC*) [9]. These parameters are potentially sensitive to local variations in water proton mobility resulting mainly from modification of water–macromolecule interactions and changes in the matrix structure [10]. The study of *T*_1_ and/or *T*_2_ can be used to assess food microstructure [11,12]. *T*_1_ and *T*_2_ calculations must be carried out by fitting the signal curves derived from the nuclear magnetic resonance (NMR) experiment to appropriate exponential equations. In the simplest case, these equations might depend on a single relaxation time, thus the fitting results in one *T*_1_ or *T*_2_ for the whole sample or Region of Interest (ROI) [9]. However, many food matrices are complex systems, thus implying a dependence of the observed NMR and MRI signal(s) on a mixture of relaxation time values, coming from different structures and chemical compounds. Therefore, it is possible to fit the curves to equations that depend on two or more exponential factors; furthermore, it is possible to obtain a continuous distribution of relaxation times utilizing Inverse Laplace Transformation [13]. Remarkably, these valuable data could be collected in real time, as current portable and cost-effective instruments on the market already exist to improve in-line product monitoring [10,11]. In addition, the monoexponential approach would give mean values that can be easily related to physicochemical and textural properties associated with food microstructure.

In the present study, cheese varieties from two geographical locations (CLM and CL) manufactured with two different procedures (T and I) were considered. This study aimed to monitor the structural changes in four cheese varieties during ripening time (RT) by MRI and to evaluate the potential of this technique as a tool to estimate physicochemical and textural properties.

Considering the stated purpose and simple approach, the monoexponential values of *T*_1_, *T*_2_, and *ADC* were obtained to relate the microstructure with the rheological behavior and to obtain an MRI map to monitor the macrostructure changes with RT. Proton density was used to analyze the hole index of the cheese matrixes.

Although advances are needed in the development of low-field equipment more suitable for the industry, the present work is a first attempt to increase the applicability of NMR to the quality control of cheese at the industrial level.

## 2. Materials and Methods

### 2.1. Experimental Design and Sample Collection

The following Spanish sheep cheese samples were categorized in this study: EMC produced industrially originating from CLM (I-CLM) or CL (I-CL), and EMC produced traditionally originating from CLM (T-CLM) or CL (T-CL). The I-CLM samples were produced in an industrial facility according to the procedure described for PDO Manchego cheese [3,14] and using pasteurized milk and recombinant Chymosin for coagulation, whereas T-CLM ones were manufactured from raw milk by a small cheesery according to an artisanal/traditional procedure using calf commercial dried rennet for enzymatic coagulation. Similarly, I-CL was produced in a large industrial facility, according to the “Castellano cheese” quality mark [4], and T-CL followed a traditional manufacturer procedure. I-CL and T-CL were made from raw milk. Recombinant Chymosin was used for the I-CL elaboration, whereas T-CL was manufactured using lamb commercial natural rennet. Homo-fermentative mesophilic LABS, as a starter culture, was added to the production of the four varieties. The processing conditions are shown in Appendix A.

Molds with a 19–20.5 cm diameter and a 14–15.5 cm height were used. Weights varied from 3.7 to 4.3 kg in fresh cheese and 3.0 to 3.5 kg at the end of ripening. Once the pieces were removed from the molds, they were salted and, finally, subjected to a period of maturation (up to 180 d). Twenty-five cheese samples from each EMC variety were analyzed throughout the ripening process at 2, 9, 30, 90, and 180 d after production (five different samples from each variety were analyzed at each sampling time). A separation distance of 2 cm from both the outer perimeter and the center was maintained for sampling (Appendix A).

### 2.2. Physicochemical Analysis

*a_w_* was measured using a Decagon CX1 hygrometer (Decagon Devices Inc., Pullman, WA, USA) at 25 °C. The pH was determined in a homogenate of the sample with distilled water (1:10 *w*/*v*), using a Digit-501 pH meter (Crison Instruments LTD, Barcelona, Spain). The protein and dry matter (DM) contents were determined using the methods of AOAC (2005) [15]. The fat content was determined using the method described by Hanson and Olley [16]. Water content (WC) was established as 100 DM.

### 2.3. Textural Analysis

Texture profile analysis (TPA) was performed at 25 °C using a TAXT2i SMS Stable Micro Systems Texture Analyzer (Stable Microsystems Ltd., Godalming, UK) with the Texture Expert program. A double compression cycle test was performed of up to 50% compression of the original portion height in cylinders (1 cm high × 1.5 cm wide) sampled with an aluminum cylinder probe P/25. A time of 5 s was allowed to elapse between the two compression cycles. Force–time deformation curves were obtained with a 30 kg load cell applied at a crosshead speed of 2 mm/s. TPA parameters were calculated according to Romero de Ávila et al. [17].

### 2.4. Magnetic Resonance Imaging Analyses

The MRI experiments were performed in a Bruker BIOSPEC 47/40 spectrometer (Bruker GmBH, Ettlingen, Germany) operating at 4.7 T equipped with a 6 cm inner-diameter gradient system at 18 °C. Two portions of each cheese (50 ± 2 g) were sampled (Appendix A) at each RT. Each sample was cut 4 cm long, 3 cm wide, and 0–1.5 cm thick. Then, the samples were placed in a 3.5 cm inner-diameter volume radiofrequency coil.

For measurements of *T*_2_ values, a multi-echo Carr–Purcell–Meiboom–Gill (CPMG) spin-echo sequence was used (60 echo series generated). In every phase step, each echo was used to obtain an image with a different echo time (TE). The TE varied from 5 ms to 300 ms. This TE interval was used to cover a wide range of *T*_2_ and to ensure the complete fall of the echo train signal. Other imaging parameters were repetition time (TR) = 5300 ms; number of averaged experiments (NAs) = 2; field of view (FOV) = 7 × 3.5 cm^2^; slice thickness = 1 mm; number of slices = 1; and matrix size = 256 × 128.

For *T*_1_ calculation, a spin-echo sequence with variable TR was considered. A series of twelve spin-echo images were acquired with logarithmically spaced TR (52.5–6002.5 ms) and constant TE (5 ms). The geometrical imaging parameters used were the same as those used for the *T*_2_ calculation.

To measure *ADC*, a series of spin-echo images was acquired at six diffusion weightings. The diffusion gradient duration (δ) was 10 ms and the time between gradients (Δ) was 40 ms. The gradient strengths varied from 5 to 450 mT/m for the first RT (2, 9, and 30 d), so the b-factor varied from 6.56 to 53,139.33 s/mm^2^. The maximum gradient strength was increased to 750 mT/m to measure the low *ADC* of the last RT (90 and 180 d), with the b-factor used being 147,609.25 s/mm^2^. The geometrical imaging parameters used were the same as those used for the *T*_2_ calculation.

For the calculation of the parametric maps and their analysis, the ImageJ 1.52a (Wayne Rasband, NIH, Bethesda, MD, USA) software was used. The MRI Analysis Calculator plugin by Karl Schmidt was used to calculate *T*_2_, *T*_1_, and *ADC* quantitative maps. The signals were fitted according to the equations described by Herrero et al. [9] but, previously, the pixels of the spin-echo images where the signal was under a certain threshold (similar signal to the background one) were considered as holes filled with air. The image signal of these areas was set to 0, so the *T*_2_, *T*_1_, and *ADC* calculation algorithms resulted in NaN (not a number) and, therefore, were excluded from the mean value calculations. After that, five ROIs were defined for each sample and the mean and the standard deviation (SD) of each parameter were calculated for each ROI. These ROIs were carefully chosen to avoid the sample edges or areas with artifacts.

For the hole index calculation, proton density (PD) images (TR/TE = 5300/5 ms, spin-echo sequence) were used, i.e., the first echo of the multi-echo series used for the *T*_2_ calculation. In these images, pixels with a signal under a certain threshold (similar signal to the background signal) were considered holes filled with air and set to 0. The number of pixels with a signal value set to 0 was compared to the total number of sample pixels.

### 2.5. Statistical Analysis

Statistical analyses were conducted using SAS 9.4 [18]. The effects of the cheese varieties (four levels: I-CL, T-CL, I-CLM, and T-CLM; two geographical origins: CLM and CL; and two manufacturing procedures: I and T) and the RT (five levels: 2, 9, 30, 90, and 180 d) on the physicochemical, MRI, and textural parameters were analyzed as a factorial experimental design (2 × 2 × 5). The physicochemical determinations were carried out in triplicate. The Shapiro–Wilk test was applied to check data normal distribution fitting. Two-way ANOVA analysis was used to determine the simultaneous effects produced by variables and Duncan’s test was performed for multiple mean comparisons. Data were reported as the mean and the SD of each EMC and the root mean square error (RMSE) and *p*-value defined the statistical analysis.

Linear and non-linear regression models were performed to determine the relationships between MRI parameters and RT, WC, *a_w_*, and TPA parameters, including the natural logarithm of the physicochemical and textural parameters and the MRI values as shown in the following expression: $\ln Y=\beta_{0}+\beta_{1}\ln X_{1}+\beta_{2}\ln X_{2}+\beta_{12}\ln X_{1}\ln X_{2}$, where *In Y* is the predicted response (natural logarithm of the estimated variable), and *β*_1_, *β*_2_, and *β*_12_ are the coefficients estimated from regression and represent the linear and cross-product effects of *ln X*_1_ and *ln X*_2_ (MRI parameters: *T*_1_ and *T*_2_) on the response.

Variables were selected in the multivariate model using the backward (stepwise) elimination procedure [19]. A significance level of *p* < 0.05 was applied for variable retention. The accuracy of prediction was evaluated in terms of the coefficient of determination (R^2^) and RMSE.

Because EMCs are heterogeneous products at a macroscopic scale, three samples of each cheese were taken from various locations for physicochemical analysis. In the case of the TPA parameter determination, six samples were taken from each half of the cheese. However, two samples from a different part of each cheese were considered for MRI studies (Appendix A).

## 3. Results and Discussion

### 3.1. Physicochemical Parameters

The statistical analyses of the effect of the EMC variety and RT on the physicochemical characteristics (Appendix A) are shown in Table 1. EMC (*p* < 0.0001) and RT (*p* < 0.0001) affected the pH. On average, CLM and I-EMC varieties had a higher pH of 0.21 units compared to CL and T-EMC, respectively (Appendix A). A statistically significant triple interaction (*p* < 0.0001) between RT and EMC varieties was found (Table 1). Similar values were observed after 30 d of ripening for T-CLM and T-CL, whereas 7.0% and 2.3% decreases in pH values during the first week were detected for T-CLM and T-CL, respectively. I-CLM and I-CL did not show clear differences (Table 1 and Appendix A). These pH values agreed with those of prior studies for Manchego cheese [20,21]. Accordingly, Ferrazza Fresno et al. [22] reported similar pH values for EMC from the same geographical origin. Medina and Núñez [1] reported that, in EMC, pH decreased in the artisanal cheeses within 24 h after manufacturing, whereas in pasteurized milk cheeses, a decrease in pH occurred during the first week. In CL cheeses, made from raw milk, the drop in pH values would have occurred before the first sampling (2 d). Medina and Núñez [1] also stated that, since lactic acid production and pH values are directly related, a high dependence could be expected on both the present bacterial microbiota and the temperature control during manufacturing and ripening, both influencing the lactose hydrolysis rate. Regarding *a_w_*, values decreased along with ripening (*p* < 0.0001). A significant interaction was found between EMC varieties and RT (*p* < 0.0001, Table 1). The EMC from CL showed a higher decrease in *a_w_* than those from CLM (6.9% vs. 4.0%, respectively) (Appendix A). Again, the obtained results agreed with previous data for a similar type of EMC [22,23]. Values of WC decreased as the RT progressed (*p* < 0.0001), although the decrease rate was higher in the I-EMC (31.8%) compared to T-EMC (21.8%) (*p* < 0.0001; Table 1 and Appendix A). A progressive water loss during the RT is a well-known process and is directly dependent on the proteolysis rate and, indirectly, on the pH changes and/or the bacterial microbiota [24]. In this case, the use of recombinant Chymosin rennet might imply both quicker milk coagulation and higher proteolysis, thus explaining I-EMC behavior. Lower fat content was detected in I-CLM compared to the others (51.4% vs. 54.2%, respectively; *p* < 0.0001 for EMC effect) but no statistically significant changes were detected with RT. Protein content was characterized by a statistically significant interaction between RT and type of manufacture (I or T). T-EMC showed an increase of 9.7% while I-EMC showed an increase of 3.7% (Table 1 and Appendix A). These findings agreed with the data reported by Revilla et al. [25].

### 3.2. Study of the Structural Characteristics by Magnetic Resonance Imaging: T_1_ and T_2_

Hyperintense areas in *T*_2_ maps (Figure 1) corresponded to bulk water (*T*_2_ > 80 ms at 4.7 T). The dehydration and free water loss led to a decrease in the transverse relaxation time and, therefore, to hypointense zones in the *T*_2_ maps (*T*_2_ < 30 ms), whereas areas with intermediate values of *T*_2_ (between 30 and 80 ms) were displayed as zones of intermediate intensity corresponded to free fat. The dark areas (set to 0 ms) corresponded to porous structures or cavities containing air.

Keeping in mind that the cheese matrices are very heterogeneous and, therefore, high SD values are shown, the hole index (%) varied (average values) from 1% to 5% for CLM and from 2% to 12% for CL (Table 2 and Appendix A). Such a high SD was shown in both I and T-EMC. No clear relationship associated with RT was found. However, the I-CLM matrix might be more compact than that of the rest of the cheeses. In Figure 1, it can be observed that CL cheese varieties showed a less compact structure than CLM. Moreover, the image of PD (Appendix A) facilitates the visualization of such a structural feature. Consequently, CLM matured for 180 d revealed widely spaced and smaller eyes than CL matrices, which revealed holes that maintained or increased size throughout the RT. In fact, CL PD images (Appendix A) showed large openings and holes (eyes) interspersed throughout the matrix, being more defined in I-CL, showing a cheese matrix/hole ratio between 10 and 18, whilst a ratio between 18 and 47 was observed for T-CL. As mentioned, the SD for the cheese matrix/hole ratio could be related to the heterogeneity of the structure, and the mean values would correspond with the porosity of each cheese variety. Concurrently, I-CLM and T-CLM showed cheese matrix/hole ratios ranging between 21 and 68, thus implying that I-CLM showed the most compact matrix (Table 2).

MRI is a feasible technology to analyze cheese porosity, which is related to the development of eyes. Eye growth in semi-hard cheese varieties requires the presence of nuclei (microscopic bubbles in the curd), adequate development of propionic bacteria for gas production (dependent on rennet composition and bacteria distribution), and an appropriate curd texture [26]. These authors indicated that the eye growth gradient could be related to the presence of micro-eyes and micro-cracks during the initial stages of ripening that function as fragility zones in the cheese matrix, thus allowing further development of bigger holes and matrix restructuration. The gas found in such junctions would be a result of the cheese manufacturing process [26]. In our study, the differences among the studied cheese varieties could be related to the different manufacturing procedures for curd development.

Moreover, differences in bacteria repartition could participate in the opening gradient. In Emmental, a semi-hard variety in which anisotropy has been found in curd grain organization, each step of the production process had an impact on the microstructure: fat globules lost their native globule aspect and organized themselves into clusters before ripening, and the protein network lost its micelle organization, becoming a continuous network after pressing [26,27].

Focusing on the continuous cheese matrix, especially at 180 d RT, a hypointense mesh/grid was observed (Figure 1 and Appendix A). This mesh/grid showed lower *T*_2_ values (at 180 d RT, 19 ms for T-CLM, 27 ms for I-CLM and I-CL, and 49 ms for T-CL) surrounding areas with higher *T*_2_ values (at 180 d RT, 46–50 ms for CLM and 80 ms for CL) (Figure 1). The areas characterized by lower *T*_2_ values could be related to a more compact protein matrix, with higher protein–protein interactions and partially dehydrated, while the areas with higher *T*_2_ values could be related to a less compact protein matrix that withholds water and/or fat (Figure 1 and Appendix A). Such microstructure changes could be related to the transformation to a fibrous casein matrix during the RT described by Everett and Auty [11], and the proteolysis and dehydration evolution in Mozzarella cheese [28].

*T*_2_ mean values of CLM showed the opposite behavior to CL values throughout the RT (Table 3). *T*_2_ CLM values decreased (27.0% for I-CLM and 39.1% for T-CLM), whereas *T*_2_ CL values increased (37.9% for I-CL and 67.0% for T-CL) (*p* < 0.0001 for the interaction; Table 1 and Table 3). CLM showed the highest *T*_2_ values from 2 to 9 d of ripening. These findings suggested that the young CLM matrices should present bigger spaces filled with bulk water than the correspondent CL ones (lower *T*_2_). At the end of ripening (90 to 180 d), CL showed higher *T*_2_ values than CLM. It has been described that higher *T*_2_ values would correspond to bulk water or fat that would be filling in the pores or spaces (pools) within the matrix [29]. A reduction in the porosity, the amount of free water, or an increase in the protein–water binding would produce a drop in *T*_2_ value [30].

So, for CL, the casein matrix evolved from a uniform structure with abundant water tightly bound to the protein matrix (low *T*_2_, 30–45 ms at 4.7 T) to a porous structure with free-flowing fat (45–80 ms) that became partly dehydrated throughout the ripening process. On the contrary, for CLM, the ripening process led to a greater increase in protein–water, protein–protein, and protein–fat interactions, thus closing the cheese matrix structure, and maintaining or reducing the percentage of holes (Table 2, Figure 1 and Appendix A). As a result, the CLM matrix should be characterized by higher water-to-casein interactions and lower free fat formation, thus leading to a lower *T*_2_ value.

The differences among ECM varieties may be attributed to the variations in the manufacturing processes, mainly to milk pre-treatment (raw vs. pasteurized), milk microbiota, and type of rennet [24]. Pasteurization influences the biochemistry of cheese ripening by altering the indigenous milk microflora, partially or completely inactivating certain indigenous enzymes, and by slight denaturation of whey proteins [31]. Supportively, Mariette [32] explained that the regular sphere-shaped holes were a consequence of the production of CO_2_ from the biochemical activity of the bacteria, whereas the irregularly shaped holes were explained by mechanical constraints during curd manipulation.

*T*_1_ is a measure of molecule mobility, representing the binding of water protons, mainly to macromolecules [9,33]. As observed in the *T*_1_ maps (Figure 2), the higher *T*_1_ values (400–800 ms, hyperintense areas) corresponded to higher WC tissues. Thus, as the WC decreased throughout the RT, *T*_1_ values also decreased [9]. In addition, as the dehydration progressed, the contribution of fat molecules had more weight in the mean *T*_1_ value, which further contributed to the *T*_1_ decrease (*T*_1_ < 400 ms).

As in the *T*_2_ maps, a relaxation time anisotropy was also observed in the *T*_1_ maps (Figure 2). The *T*_1_ maps suggested a more uniform matrix in the CLM structure than in CL. But, overall, the cheese matrices were characterized by interspersed holes containing air, fat clusters (*T*_1_ < 400 ms), and a fine hydrated network (hyperintense mesh with *T*_1_ values at 180 d with an upper limit of 400–500 ms) over a hypointense background (the most compact and dehydrated matrix, with *T*_1_ values at 180 d between 300 and 350 ms). These areas could be related to matrix compaction as a result of more or stronger protein–protein interactions and/or entrapped fat.

As shown in Table 3, in the four EMC varieties, as the RT increased, the *T*_1_ value decreased. As in the case of *T*_2_, a significant interaction of the studied factors (EMC variety and RT) was found (*p* < 0.0001; Table 1). CL showed higher *T*_1_ values than CLM. Since *T*_1_ is a measure of the effect of the external environment on the spins, these results suggested that a freer distribution of both water and fat molecules would characterize the matrix of CL compared to CLM.

### 3.3. Texture Profile

Statistically significant differences between EMC variety and RT were found for all the TPA parameters (*p* < 0.0001; Table 1). The results showed that throughout ripening, the highest values of (*p* < 0.0001) both hardness (from 22.2 N to 42.1 N) and gumminess (between 13.7 N and 16.1 N) corresponded to I-CLM (Table 4), which could be due to the more compact and uniform structure of this cheese variety as previously discussed. This fact may be related to the use of pasteurized milk and to manufacturing aspects (Appendix A), such as curd pressing time, starter culture, and type of rennet [20]. Núñez [24] indicated that Manchego cheeses elaborated with pasteurized milk exhibit a firmer texture than those made from raw milk. Higher proteolysis has been found in raw milk cheese, thus weakening the protein network and reducing firmness [23,34]. Overall, a hardness increase with RT, associated with a parallel WC decrease, was observed (Table 4). Accordingly, Lobato-Calleros et al. [25] and Revilla et al. [35] stated that, with the WC decrease, an increase in protein–protein interactions must be considered, thus allowing higher complexity of the cross-linking of the 3D structure. Gumminess values showed the opposite behavior to hardness, in which gumminess decreased as RT increased. Matrices older than 30 days showed the highest adhesiveness and springiness values (*p* < 0.05; Table 4). T-CL and T-CLM showed higher adhesiveness than those of industrial manufacturing at the end of RT. In agreement with the results, increasing amounts of free fat might imply a more adhesive cheese surface. In general, the cohesiveness decreased as the RT increased. This phenomenon may be related to the progressive dehydration of the cheese matrix with ripening, becoming brittle, and reducing its resistance to deformation. T-CL showed the lowest cohesiveness values, likely because this cheese variety had a more open structure than the other EMC at longer RT (Table 2). Regarding chewiness, young pieces (RT of 2 to 9 d after manufacturing) showed lower values than the longer ripened ones.

Similar changes in textural features during RT were described for Manchego cheese [31], Terrincho ewe cheese [36], and Cheddar cheese [37]. Núñez [24] reported that higher values of hardness and springiness were found in raw and pasteurized milk cheeses when made with autochthonous strains. Lobato-Calleros et al. [35] described changes in chewiness and adhesiveness associated with milk fat. González-Viñas et al. [31] described more homogeneous textural characteristics for industrial Manchego cheese varieties than for their artisanal counterparts. Therefore, the factors (Appendix A) that take part in the manufacturing of the studied cheese varieties, such as starter culture and rennet, would explain the differences among the values of the texture features.

### 3.4. Study of the Relationship between T_1_/T_2_ and Physicochemical and TPA Parameters: Predictive Models (Industry Applicability)

Regression models were used to determine the degree of association of *T*_1_ and/or *T*_2_ with RT, WC, and *a_w_* to provide insight into the relationship between these parameters and their potential use in industry applications (Table 5 and Appendix A). Together, imitating the tendency of a previous study [8], linear regressions were also calculated individually for each MRI parameter and each EMC. Equations with higher R^2^ values were detected when considering the logarithmic approach compared to the linear individual ones. Higher R^2^ values were obtained for T-EMC models than for I-EMC with both *T*_1_ and *T*_2_ parameters (Table 5). Regarding the R^2^ values, in the case of linear models, the lowest value was observed for I-CL for *T*_1_ for the considered parameters (RT, 0.39; *a_w_*, 0.39; and WC, 0.56) whereas, for *T*_2_, the lowest value was shown in I-CLM (RT, 0.35; *a_w_*, 0.49; and WC, 0.53). However, when observing the logarithmic models, values of R^2^ ≥ 0.70 were observed for both *T*_1_ and *T*_2_. The *T*_2_ linear models showed higher absolute values for slope compared to *T*_1_. In the case of *T*_1_, all the slope values of the linear models resulted in negative values for RT and positive values for *a_w_* and WC. But, in the case of *T*_2_, an opposite behavior between CLM and CL was observed (Appendix A).

For a better understanding of the matrix microstructure and the textural response, the relationship between textural and MRI parameters was studied (Table 5 and Appendix A). Together with the logarithmic models, the development of linear regressions was also carried out individually for each MRI parameter and each EMC [8]. Again, logarithmic models showed higher R^2^ values compared to linear ones. Moreover, paying attention to logarithmic models of hardness, adhesiveness, and cohesiveness (Table 5 and Appendix A), T-EMC showed higher R^2^ values than I-EMC. Consequently, this would imply that the changes in the traditional matrices, being more heterogeneous (greater size and more irregular eye distribution), are well represented by the *T*_1_ and *T*_2_ parameters.

This study demonstrates that the monitoring of the physicochemical characteristics of cheese can be carried out by measuring *T*_1_ and *T*_2_ variables, using logarithmic regression models for the considered cheese varieties. Nevertheless, a regression model must be individually obtained for each cheese variety. Together, statistical differences were detected either considering the manufacturing location (CLM vs. CL) or the manufacturing process (I vs. T), thus implying the suitability of MRI as a technique to control Spanish sheep cheese adulterations. Lerma-García et al. [38] reported similar results when using Fourier-transform infrared spectroscopy (FTIR). Nevertheless, further studies would be necessary to fully interpret *T*_1_ and *T*_2_ behavior.

### 3.5. Apparent Diffusion Coefficient (ADC)

The average translational motion of water protons could be quantified by *ADC* values, thus indicating the magnitude of the diffusion of water molecules through a biological matrix [39,40]. Some barriers, such as cell walls and large-chain proteins, reduce the ability of water to diffuse [9]. Therefore, *ADC* maps may enable the assessment of changes in spin diffusion through cheese matrices throughout the RT.

Regarding cheese matrices, in the *ADC* maps (Figure 3), hyperintense areas corresponded to a higher diffusion of water molecules through the matrix (*ADC* > 3.5 × 10^−5^ mm^2^/s). In this case, the higher the RT considered, the lower the *ADC* values, revealing hypointense (*ADC* < 0.45 × 10^−5^ mm^2^/s) maps at 180 d. The changes observed in Figure 3 reflect both the water loss and the consolidation of the cheese structure, both of which are related to a decrease in the *ADC* values (Table 3). Prior studies have described that lower *ADC* values are associated with denser tissues [39].

The decrease in *ADC* values (Table 1 and Table 3; *p* < 0.0001) is associated with the decrease in the WC throughout the RT but also with the formation of protein–protein and protein–water interactions as the cheese matrix stabilizes, particularly during the first month of ripening. In this microenvironment, water mobility was more restricted. Moreover, such a decrease in *ADC* values can be also related to the increase in relative fat content due to the decrease in WC with RT (Appendix A), as previously observed in fatty livers [41] and model food products [42]. It has been suggested that the presence of fat may decrease *ADC* values by restricting the diffusion of water [41]. Although fat diffusion happens, being more feasible in matrices with higher dehydration, fat *ADC* values are much lower than water values [43], thus justifying higher *ADC* values of hydrated matrices. Steidle et al. [43] described that fat *ADC* values were approximately two orders of magnitude smaller than water values due to the higher molecular weight of triglycerides, thus explaining that an *ADC* value increase was not observed in the CL cheese variety, whose *T*_2_ value increase with RT was related to the presence of a higher number of free fat protons.

A lower decrease rate of *ADC* values was detected for T-EMC than for I-EMC (55.8% vs. 67.2%, respectively; *p* < 0.01), which could be related to a more open structure with larger eyes. It appears that not only *ADC* absolute values but also differences in RT should be considered when relating to matrix microstructure. In the protein structure of meat systems containing plasma powder, significant correlations between *ADC* values and WC have been previously reported [9]. Regarding the regression models between RT, WC, and *a_w_* of EMC and *ADC*, although statistically significant (*p* < 0.05), R^2^ values of the linear regression were lower than 0.60 for RT (between 0.42 and 0.54); for WC lower for T-EMC (0.47 and 0.58 for CLM and CL, respectively) than for I-EMC (0.73 for CLM and 0.69 for CL); and for *a_w_* between 0.42 and 0.67. Overall, lower R^2^ values than those obtained for *T*_1_ and *T*_2_ were obtained, which could be related to the extremely low *ADC* values at the end of the RT that could be related to fat interference (Appendix A).

## 4. Conclusions

Magnetic Resonance Imaging (MRI) provided valuable information about the matrix structure of sheep cheese, which can be related to differences in the geographical origin, manufacturing process (industrial vs. traditional), milk pre-treatment (raw vs. pasteurized), and ripening time.

*T*_2_ values were sensitive to both the porosity of the cheese matrix and the water linked to macromolecules. Therefore, *T*_2_, together with proton density images, were suitable for monitoring structural modifications of the cheese matrix during ripening. *T*_1_ maps proved to serve as a useful tool to monitor the ripening process of the sheep cheese, depending on the surrounding molecules of fat and water. *ADC* maps assessed the diffusion changes throughout the ripening time by giving attention to the water losses of the cheese matrix, thus implying a decrease in the *ADC* values and the consolidation of the structure. High values of regression coefficients were detected for logarithmic estimation models of water content, *a_w_*, ripening time, hardness, adhesiveness, springiness, and cohesiveness depending on *T*_2_ and *T*_1_ values. However, further studies are necessary for individual optimization of the model equations for each product and production procedure. Therefore, these results support the suitability of MRI, an emerging non-destructive technique, in the monitoring of cheese structure changes during ripening, the estimation of the physicochemical and textural-related properties, and the analysis of the basis of the structural characterization of different varieties of sheep cheese. These promising results could become part of the NMR technology transfer to the manufacturing control in industrial processing lines.

## Figures and Tables

**Figure 1 foods-13-03225-f001:**
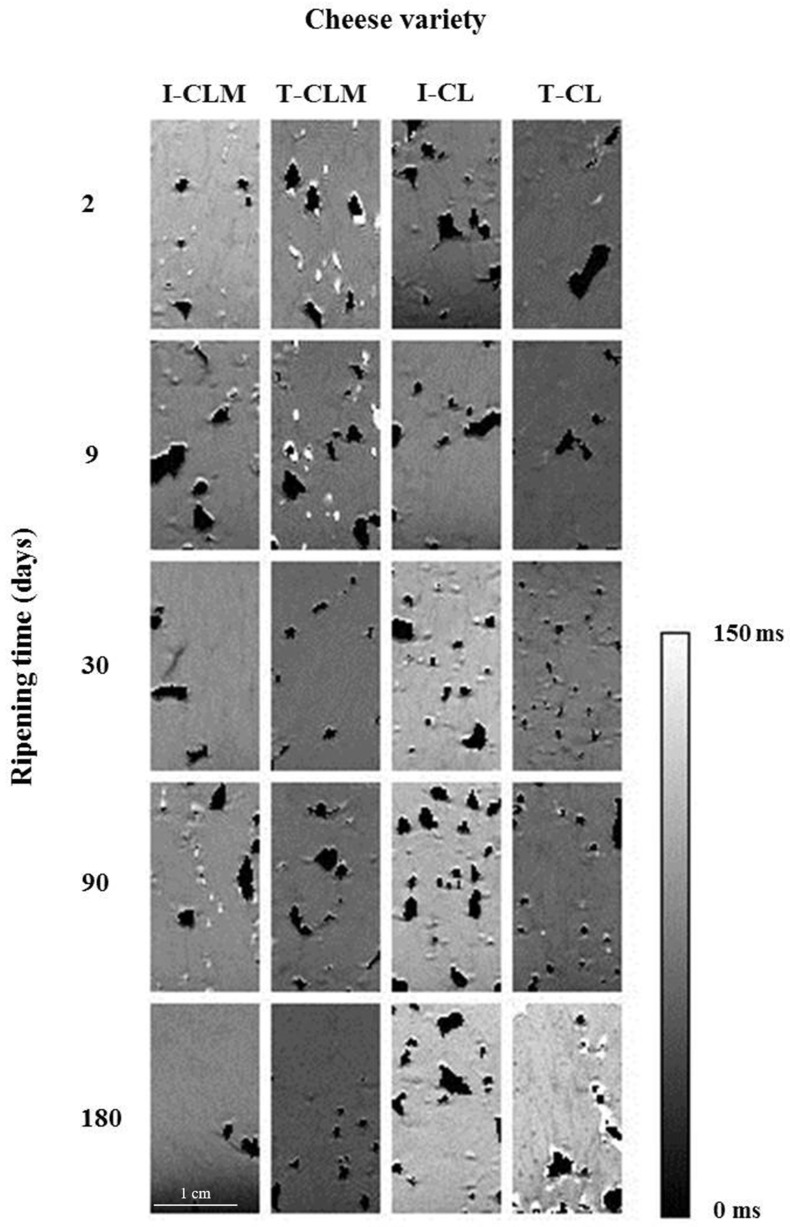
Representative *T*_2_ maps of the four varieties of Spanish sheep cheese at different ripening times. Cheese variety (I-CLM, T-CLM, I-CL, and T-CL) according to Table 1. A scale bar (zoomed in) corresponding to 1 cm is shown.

**Figure 2 foods-13-03225-f002:**
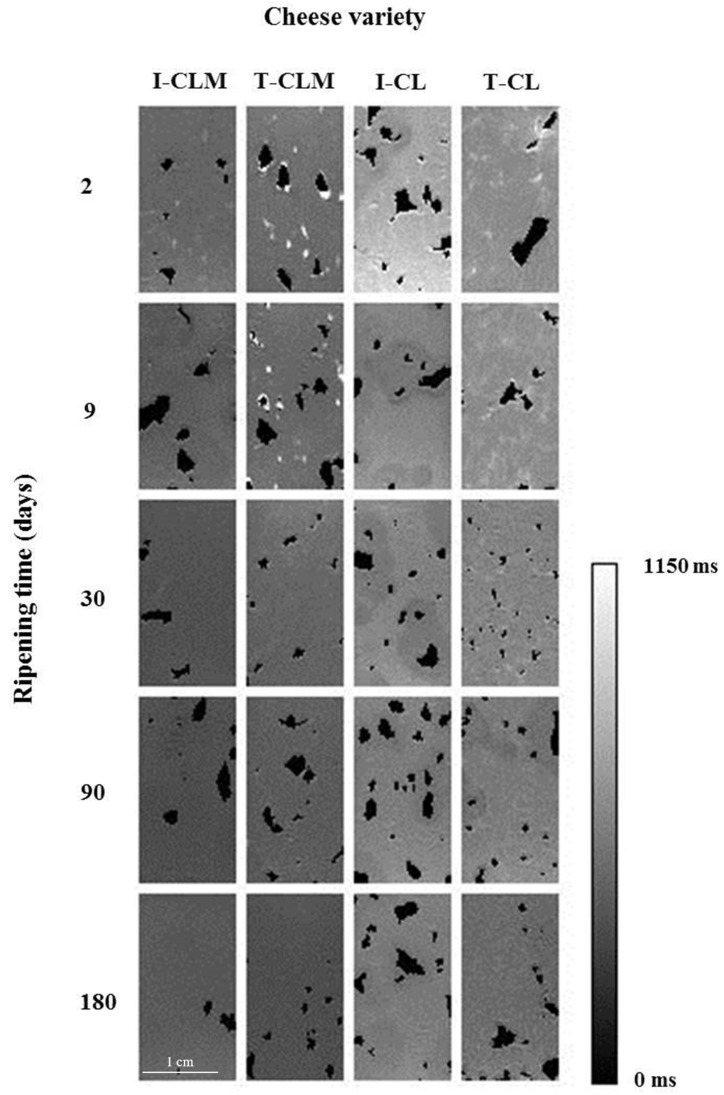
Representative *T*_1_ maps of the four varieties of Spanish sheep cheese at different ripening times. Cheese varieties (I-CLM, T-CLM, I-CL, and T-CL) according to Table 1. A scale bar (zoomed in) corresponding to 1 cm is shown.

**Figure 3 foods-13-03225-f003:**
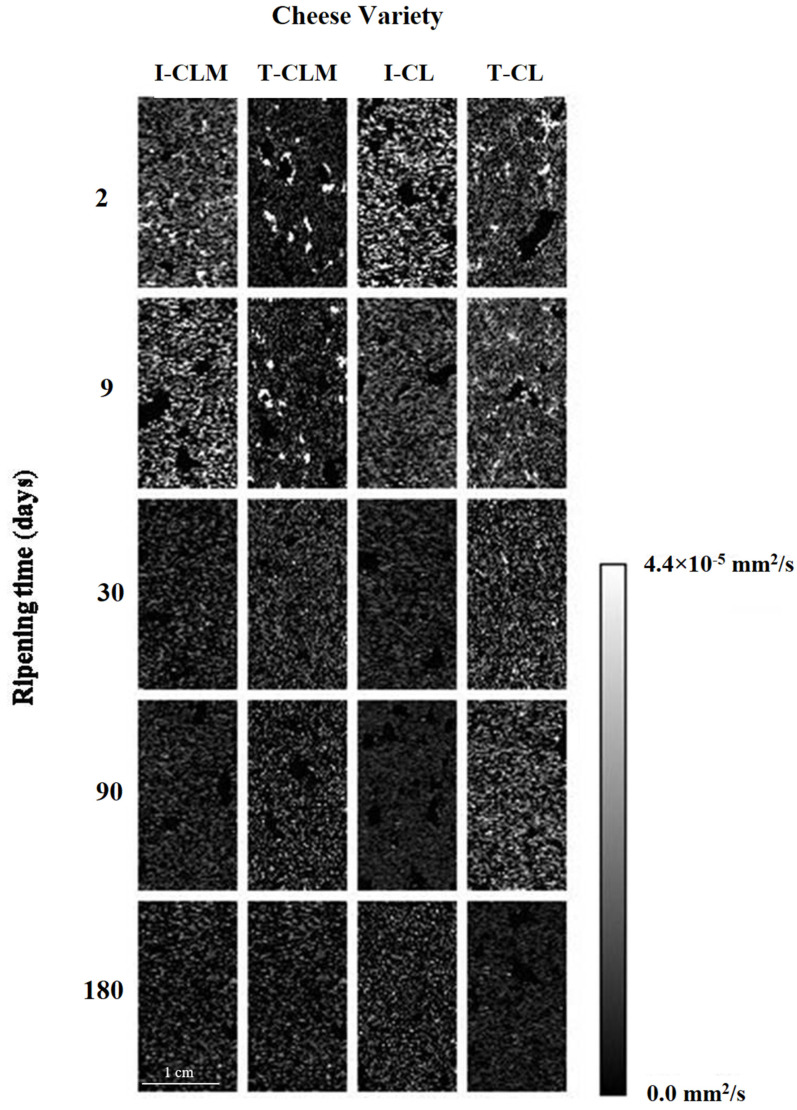
Representative *ADC* maps of the four varieties of Spanish sheep cheese at different ripening times. Cheese varieties (I-CLM, T-CLM, I-CL, and T-CL) according to Table 1. A scale bar (zoomed in) corresponding to 1 cm is shown.

**Table 1 foods-13-03225-t001:** Effect of sheep cheese variety geographical origin (A: CLM and CL), manufacturing procedure (B: I and T), and ripening time (C: RT) on physicochemical, magnetic resonance, and texture parameters.

		RMSE ^a^	*p*-Value ^b^						
		(n = 25)	A	B	C	A × B	A × C	B × C	A × B × C
Physicochemical									
	pH	0.0527	0.0001	0.0001	0.0001	0.0001	0.0001	0.0001	0.0377
	Protein (% DM ^c^)	1.3738	0.0801	0.0787	0.2010	0.1166	0.4237	0.0131	0.4340
	Fat (% DM)	1.7407	0.0001	0.0381	0.9146	0.0147	0.4680	0.7840	0.8564
	Ash (% DM)	0.3488	0.0001	0.0130	0.0001	0.0010	0.1132	0.1051	0.0175
	WC ^d^ (%)	1.3294	0.0001	0.0001	0.0001	0.7949	0.0908	0.0001	0.1221
	*a_w_*	0.0050	0.0001	0.1975	0.0001	0.8469	0.0001	0.0001	0.0001
Structural									
	*T*_1_ (ms)	14.679	0.0001	0.0001	0.0001	0.0001	0.0001	0.0001	0.0001
	*T*_2_ (ms)	1.8984	0.0001	0.0001	0.0001	0.0001	0.0001	0.0001	0.0001
	*ADC* (mm^2^/s)	0.0015	0.3112	0.0004	0.0001	0.0022	0.0057	0.0001	0.0110
Textural									
	Hardness (N)	3.9115	0.0001	0.1028	0.0001	0.0001	0.1099	0.0001	0.0021
	Adhesiveness (N × s)	0.0545	0.0155	0.0001	0.0001	0.0001	0.0001	0.0001	0.1934
	Springiness (m)	0.0004	0.0001	0.0001	0.0001	0.0001	0.0700	0.7508	0.0001
	Cohesiveness	0.0499	0.0001	0.0001	0.0001	0.0001	0.0001	0.0001	0.0001
	Gumminess (N)	1.9257	0.0001	0.0001	0.0001	0.2161	0.0001	0.0001	0.0001
	Chewiness (J)	0.0050	0.0001	0.0001	0.0001	0.5529	0.0004	0.0368	0.0097

^a^ RMSE = root mean square error; n = five different pieces of each variety were analyzed at each sampling time. ^b^ A = location (CLM or CL); B = manufacture (I or T); and C = RT (2, 9, 30, 90, and 180 days). I-CLM = pieces produced in an industrial facility in the Castilla-La Mancha region; T-CLM = pieces manufactured according to an artisanal/traditional procedure in Castilla-La Mancha; I-CL = pieces produced in a large industrial facility in the Castilla y León region; and T-CL = manufactured according to an artisanal/traditional procedure in Castilla y León (Appendix A). ^c^ DM = dry matter. ^d^ WC = water content = 100 DM.

**Table 2 foods-13-03225-t002:** Hole index (% of total area) and cheese matrix to hole ratio of the four varieties of Spanish sheep cheese ^1^ at different ripening times (RTs).

	RT (Days)	I-CLM ^1^		T-CLM		I-CL		T-CL	
Hole index (%) ^2^	2	1.94 ± 0.64	bc,β	3.11 ± 0.50	ab,αβ	5.53 ± 1.43	b,α	4.01 ± 2.56	b,αβ
	9	4.79 ± 1.27	a,β	4.92 ± 2.87	a,αβ	7.52 ± 0.33	ab,α	2.89 ± 1.10	bc,β
	30	3.00 ± 1.15	ab,αβ	1.95 ± 0.61	c,αβ	5.38 ± 0.90	b,α	2.14 ± 0.38	c,β
	90	3.72 ± 0.30	a,β	4.57 ± 1.04	a,β	8.94 ± 1.51	a,α	6.25 ± 2.73	ab,αβ
	180	1.28 ± 0.90	c,β	2.51 ± 0.14	bc,β	7.93 ± 0.66	a,α	12.2 ± 6.1	a,α
Cheese matrix/hole ratio ^3^	2	53.7 ± 18.2	a,α	31.2 ± 8.0	b,αβ	17.8 ± 4.6	a,β	46.6 ± 16.4	a,α
	9	20.6 ± 5.7	c,β	33.6 ± 14.3	b,αβ	12.3 ± 1.6	bc,γ	37.3 ± 14.9	ab,α
	30	35.0 ± 11.8	b,αβ	56.2 ± 22.2	a,α	17.9 ± 3.1	a,β	46.7 ± 8.6	a,α
	90	26.0 ± 2.1	bc,α	21.6 ± 5.2	c,αβ	10.4 ± 2.1	bc,β	17.8 ± 9.8	b,β
	180	68.1 ± 10.1	a,α	38.9 ± 2.3	a,β	11.7 ± 1.0	c,γ	24.0 ± 17.0	b,βγ

a, b, c: Different letters within a column for the same textural characteristic indicate statistically significant differences (*p* < 0.05) related to ripening time. α, β, γ: Different letters within a row indicate statistically significant differences (*p* < 0.05) related to cheese variety. ^1^ Cheese variety (I-CLM, T-CLM, I-CL, and T-CL) according to Table 1. ^2^ Hole index (%): area percentage corresponding to holes. ^3^ Cheese matrix to hole ratio (matrix area/hole area).

**Table 3 foods-13-03225-t003:** Magnetic Resonance Imaging parameters (mean values) of the four varieties of Spanish sheep cheese at different ripening times (RTs).

	RT (Days)	Cheese Variety ^1^							
		I-CLM		T-CLM		I-CL		T-CL	
*T*_2_ (ms)	2	59.6 ± 1.5	a,α	45.3 ± 0.8	a,β	37.5 ± 3.6	d,γ	21.0 ± 0.7	c,δ
	9	47.0 ± 1.6	b,α	37.6 ± 1.1	b,γ	42.6 ± 1.3	c,β	22.2 ± 0.8	c,δ
	30	45.2 ± 1.1	c,β	36.1 ± 0.7	b,γ	58.2 ± 1.4	b,α	36.4 ± 1.6	b,γ
	90	44.1 ± 0.5	cd,β	34.6 ± 1.8	b,γ	57.9 ± 1.2	b,α	35.2 ± 1.5	b,γ
	180	43.5 ± 1.1	d,β	27.6 ± 0.7	c,γ	60.4 ± 1.6	a,α	63.7 ± 1.8	a,α
*T*_1_ (ms)	2	473.3 ± 12.3	a,β	499.1 ± 15.7	a,β	666.8 ± 15.6	a,α	662.0 ± 18.8	a,α
	9	424.3 ± 15.4	a,δ	486.4 ± 11.4	a,γ	597.1 ± 16.7	b,β	659.6 ± 19.0	a,α
	30	381.1 ± 9.6	b,δ	462.8 ± 9.3	b,γ	574.4 ± 15.3	c,β	602.0 ± 18.5	b,α
	90	355.4 ± 9.9	c,γ	424.7 ± 9.5	c,β	569.5 ± 11.6	cd,α	549.4 ± 20.1	c,α
	180	362.4 ± 7.3	c,δ	404.1 ± 5.4	d,γ	557.9 ± 5.7	d,α	485.8 ± 10.2	d,β
*ADC *^2^ (mm^2^/s, ×10^−5^)	2	1.05 ± 0.14	a,α	0.92 ± 0.13	a,α	1.09 ± 0.23	a,α	0.89 ± 0.13	a,α
	9	0.96 ± 0.11	a,α	0.88 ± 0.11	b,α	0.75 ± 0.20	b,α	0.94 ± 0.13	a,α
	30	0.40 ± 0.08	b,β	0.55 ± 0.14	c,αβ	0.41 ± 0.11	c,β	0.74 ± 0.21	b,α
	90	0.39 ± 0.12	b,β	0.46 ± 0.12	c,αβ	0.37 ± 0.07	c,β	0.74 ± 0.20	b,α
	180	0.38 ± 0.13	b,α	0.43 ± 0.13	c,α	0.32 ± 0.11	c,α	0.37 ± 0.13	c,α

a, b, c, d: Different letters within a column for the same textural characteristic indicate statistically significant differences (*p* < 0.05) related to ripening time. α, β, γ, δ: Different letters within a row indicate statistically significant differences (*p* < 0.05) related to cheese variety. ^1^ Cheese variety (I-CLM, T-CLM, I-CL, and T-CL) according to Table 1. ^2^ *ADC* = apparent diffusion coefficient.

**Table 4 foods-13-03225-t004:** Texture parameters of the four varieties of Spanish sheep cheese ^1^ at different ripening times (RTs).

	RT (Days)	I-CLM		T-CLM		I-CL		T-CL	
Hardness (N)	2	22.2 ± 1.5	c,α	13.7 ± 3.3	c,γ	15.0 ± 0.8	c,γ	18.9 ± 3.0	bc,β
	9	25.8 ± 3.7	c,α	25.8 ± 3.1	b,α	17.7 ± 1.7	b,β	15.4 ± 2.7	c,β
	30	27.5 ± 6.7	bc,α	26.7 ± 4.3	ab,α	18.3 ± 1.8	b,β	21.1 ± 1.5	b,β
	90	34.1 ± 3.2	b,α	29.8 ± 6.3	a,α	21.0 ± 1.9	b,β	31.8 ± 6.6	a,α
	180	42.1 ± 4.3	a,α	31.0 ± 5.6	a,β	30.1 ± 2.6	a,β	31.2 ± 5.3	a,β
Adhesiveness (N × s)	2	−0.104 ± 0.022	c,α	−0.021 ± 0.006	c,γ	−0.047 ± 0.010	c,βγ	−0.065 ± 0.019	c,β
	9	−0.110 ± 0.047	b,α	−0.056 ± 0.029	b,β	−0.042 ± 0.004	c,γ	−0.105 ± 0.023	bc,αβ
	30	−0.129 ± 0.048	b,α	−0.070 ± 0.022	b,β	−0.077 ± 0.010	b,β	−0.142 ± 0.028	b,α
	90	−0.146 ± 0.012	b,β	−0.269 ± 0.077	a,α	−0.067 ± 0.008	b,γ	−0.325 ± 0.113	a,α
	180	−0.251 ± 0.050	a,β	−0.332 ± 0.107	a,α	−0.116 ± 0.024	a,γ	−0.315 ± 0.114	a,α
Springiness (cm)	2	0.059 ± 0.014	c,γ	0.170 ± 0.048	c,β	0.069 ± 0.006	c,γ	0.250 ± 0.035	c,α
	9	0.109 ± 0.022	b,γ	0.175 ± 0.010	c,β	0.096 ± 0.025	b,γ	0.309 ± 0.024	b,α
	30	0.116 ± 0.021	b,γ	0.193 ± 0.038	bc,β	0.107 ± 0.020	b,γ	0.328 ± 0.021	ab,α
	90	0.168 ± 0.053	a,γ	0.225 ± 0.016	b,β	0.114 ± 0.027	b,γ	0.340 ± 0.043	a,α
	180	0.160 ± 0.024	a,β	0.330 ± 0.035	a,α	0.193 ± 0.038	a,β	0.398 ± 0.044	a,α
Cohesiveness	2	0.670 ± 0.030	a,β	0.743 ± 0.037	a,α	0.661 ± 0.021	a,β	0.292 ± 0.034	a,γ
	9	0.615 ± 0.056	ab,α	0.609 ± 0.030	b,α	0.578 ± 0.066	b,α	0.248 ± 0.011	b,β
	30	0.598 ± 0.042	b,α	0.442 ± 0.034	c,β	0.466 ± 0.031	c,β	0.243 ± 0.024	b,γ
	90	0.407 ± 0.061	c,α	0.412 ± 0.038	c,α	0.343 ± 0.011	d,β	0.156 ± 0.013	c,γ
	180	0.339 ± 0.039	c,α	0.276 ± 0.033	d,β	0.273 ± 0.023	e,β	0.154 ± 0.020	c,γ
Gumminess (N)	2	14.9 ± 1.8	a,α	10.2 ± 1.9	b,β	9.90 ± 1.17	a,β	5.53 ± 1.21	a,γ
	9	15.9 ± 2.3	a,α	15.7 ± 1.8	a,α	10.2 ± 1.6	a,β	3.82 ± 0.80	b,γ
	30	16.5 ± 2.8	a,α	11.8 ± 2.3	b,β	8.53 ± 1.06	ab,β	5.14 ± 0.35	a,γ
	90	13.9 ± 3.3	b,α	12.3 ± 3.0	ab,α	7.20 ± 1.26	b,β	4.97 ± 1.18	a,γ
	180	14.3 ± 1.7	ab,α	8.56 ± 2.35	b,β	8.20 ± 1.27	b,β	4.80 ± 1.27	a,γ
Chewiness (J)	2	0.009 ± 0.003	b,β	0.017 ± 0.005	b,α	0.007 ± 0.001	b,β	0.014 ± 0.006	a,α
	9	0.017 ± 0.004	ab,β	0.028 ± 0.003	a,α	0.010 ± 0.003	b,γ	0.012 ± 0.003	a,βγ
	30	0.019 ± 0.008	a,α	0.023 ± 0.007	a,α	0.009 ± 0.002	b,β	0.017 ± 0.002	a,α
	90	0.023 ± 0.006	a,α	0.028 ± 0.006	a,α	0.008 ± 0.002	b,γ	0.017 ± 0.009	a,β
	180	0.023 ± 0.005	a,α	0.028 ± 0.006	a,α	0.016 ± 0.003	a,β	0.019 ± 0.005	a,αβ

a, b, c: Different letters within a column for the same textural characteristic texture parameter indicate statistically significant differences (*p* < 0.05) related to ripening time. α, β, γ: Different letters within a row indicate statistically significant differences (*p* < 0.05) related to cheese variety. ^1^ Cheese variety (I-CLM, T-CLM, I-CL, and T-CL) according to Table 1.

**Table 5 foods-13-03225-t005:** Statistical parameters of the regression models ^a^ considering the MRI parameters (*T*_1_ and *T*_2_), the physicochemical and texture parameters, and the ripening time (RT).

			I-CLM ^b^				T-CLM				I-CL				T-CL			
			R^2 c^	R^2^*	RMSE	*p*-Value	R^2^	R^2^*	RMSE	*p*-Value	R^2^	R^2^*	RMSE	*p*-Value	R^2^	R^2^*	RMSE	*p*-Value
RT (days) ^d^	Linear	*T* _1_	0.51	0.50	47.56	0.0001	0.74	0.73	27.36	0.0001	0.39	0.38	53.11	0.0001	0.79	0.78	22.63	0.0001
		*T* _2_	0.35	0.33	54.89	0.0001	0.76	0.75	33.6	0.0001	0.53	0.52	46.44	0.0001	0.78	0.77	21.75	0.0001
	Ln		0.93	0.92	0.452	0.0001	0.95	0.94	0.36	0.0001	0.90	0.89	0.538	0.0001	0.91	0.90	0.518	0.0001
*a_w_*	Linear	*T* _1_	0.70	0.69	0.005	0.0001	0.74	0.73	0.01	0.0001	0.39	0.37	0.02	0.0001	0.71	0.70	0.011	0.0001
		*T* _2_	0.49	0.48	0.006	0.0001	0.72	0.71	0.009	0.0001	0.57	0.56	0.062	0.0001	0.71	0.71	0.008	0.0001
	Ln		0.80	0.78	0.004	0.0001	0.94	0.94	0.006	0.0001	0.75	0.72	0.018	0.0001	0.93	0.92	0.008	0.0001
WC (%) ^d^	Linear	*T* _1_	0.70	0.69	2.88	0.0001	0.68	0.67	2.20	0.0001	0.56	0.54	4.43	0.0001	0.79	0.78	1.342	0.0001
		*T* _2_	0.53	0.52	4.41	0.0001	0.62	0.61	2.89	0.0001	0.77	0.76	2.4	0.0001	0.75	0.74	2.01	0.0001
	Ln		0.86	0.85	0.063	0.0001	0.77	0.75	0.055	0.0001	0.93	0.92	0.052	0.0001	0.94	0.94	0.025	0.0001
Hardness (N)	Linear	*T* _1_	0.51	0.50	5.81	0.0001	0.41	0.40	5.965	0.0001	0.50	0.49	3.972	0.0001	0.75	0.75	3.933	0.0001
		*T* _2_	0.44	0.43	6.199	0.0001	0.64	0.63	4.654	0.0001	0.44	0.43	4.184	0.0001	0.41	0.40	6.059	0.0001
	Ln		0.78	0.77	0.127	0.0001	0.87	0.86	0.133	0.0001	0.72	0.70	0.137	0.0001	0.87	0.86	0.122	0.0001
Adhesiveness (N × s)	Linear	*T* _1_	0.32	0.30	0.055	0.0001	0.69	0.69	0.064	0.0001	0.36	0.35	0.024	0.0001	0.69	0.68	0.074	0.0001
		*T* _2_	0.25	0.24	0.058	0.0002	0.58	0.57	0.092	0.0001	0.52	0.51	0.021	0.0001	0.40	0.39	0.102	0.0001
	Ln		0.54	0.51	0.357	0.0001	0.86	0.86	0.429	0.0001	0.71	0.69	0.223	0.0001	0.80	0.79	0.33	0.0001
Springiness (m)	Linear	*T* _1_	0.64	0.64	0.0002	0.0001	0.57	0.56	0.0004	0.0001	0.38	0.37	0.0004	0.0001	0.51	0.50	0.0004	0.0001
		*T* _2_	0.55	0.54	0.0003	0.0001	0.51	0.50	0.0004	0.0001	0.42	0.41	0.0004	0.0001	0.54	0.54	0.0004	0.0001
	Ln		0.83	0.82	0.184	0.0001	0.71	0.70	0.169	0.0001	0.67	0.64	0.261	0.0001	0.70	0.68	0.113	0.0001
Cohesiveness	Linear	*T* _1_	0.52	0.51	0.086	0.0001	0.72	0.71	0.062	0.0001	0.60	0.59	0.095	0.0001	0.72	0.72	0.025	0.0001
		*T* _2_	0.42	0.40	0.106	0.0001	0.73	0.73	0.058	0.0001	0.67	0.66	0.072	0.0001	0.51	0.50	0.042	0.0001
	Ln		0.71	0.69	0.16	0.0001	0.95	0.95	0.078	0.0001	0.75	0.73	0.175	0.0001	0.89	0.89	0.094	0.0001

^a^ Ln = logarithmic regression model according to Section 2. ^b^ Cheese variety (I-CLM, T-CLM, I-CL, and T-CL) according to Table 1. ^c^ R^2^ = coefficient of determination, R^2^* = adjusted coefficient of determination, and RMSE = root mean square error. ^d^ RT = ripening time (days); WC = water content (%).

## Data Availability

The original contributions presented in the study are included in the article/Appendix A, further inquiries can be directed to the corresponding author.

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
