# Peer review of "Magnetic Resonance Imaging (MRI) of Spanish Sheep Cheese: A Study on the Relationships between Ripening Times, Geographical Origins, Textural Parameters, and MRI Parameters"

_foods, 2024, doi:10.3390/foods13203225_

Round 1
Reviewer 1 Report
Comments and Suggestions for Authors
Although I recognize the general value of the work, I find some relevant criticisms, which are discussed in the following.
1. The MR relaxometry study is based on the oversimplifying (and wrong) assumption that magnetization decays can be described by a single component. In order for the paper to present original results, it should be inserted in the current (and widely demonstrated) context. Authors can find many works dealing with the same problem and even on very similar systems. It appears that the authors neglected a large body of evidence in discussing the MR relaxometry results. Updated data processing, which considers different pools of protons, could potentially reverse some of the conclusions.
2. I cannot see potential industrial applications of high-field MRI. The instruments are bulky and very expensive, they require highly skilled operators, require costly and expensive maintenance, data interpretation is complex and time-consuming. Whose dairy industry would be able (and interested) in investing so much on such an advanced technique? What cost-benefits ratio would you envisage in particular?
I find interesting the comparison between conventional characterizations and MRI relaxometry.
I suggest major revisions concerning MR data treatment, improvement of the references with proper literature citations, a revised discussion based on new data (different water and fat pools, for example)
Comments on the Quality of English LanguageThe English is good, just some minor adjustments could be made.
Author Response
Please, see attachment.

Reviewer 2 Report
Comments and Suggestions for Authors
This interesting study examines several different cheeses by MRI. The inclusion of Table S1 is useful and appreciated.
It can be expected that the water loss during ripening results in a more and more darkened image as can be seen in Figure S2. However, the MRI image slices in Figure S2 (proton density) do not seem to show the same slices as the ones shown for the T2 and T1 MRI images (Figure 1 and Figure 2). Why was this decision made? This makes it more difficult to draw conclusions by comparing these figures.
The values for the 'hole index' and for the 'cheese matrix / hole ratio' do not show a clear trend with respect to the ripening period. This is possibly because the data calculation relied on a insufficiently large sample size (number of images considered was too small). Cheeses are not homogeneous objects and have a lot of variations from one slice to the next.
In Table 2, Table 3, Table 4 and Table S2: Better adjust the significant figures, e.g., '3.11 ± 0.495' should be contracted to something like '3.11 ± 0.50', and '51.92±2.638' should be contracted to something like '51.9±2.6'.
Some specific issues:
Line 124: 'Two portions of each cheese (50 ± 2 g) were sampled': What was the reason for this, in particular considering that not all data were presented? In particular, Figure S2 and Figure 1 + 2 seem to show different samples.
Line 125: 'Each sample was cut 4 cm long, 3 cm wide and 1.5 cm thick': Figure S1 seems to indicate that the thickness is not uniform, should that be 0-1.5 cm thickness instead?
Lines 127-131: How many slices were measured? One? What was the name of the pulse sequence? Please verify the value that was stated for the field of view: 7 x 3.5 cm2. If that is the case, while the sample is only 3 cm wide, why do the images presented in Figure 1-3 do not have black borders 0.5 cm wide? If not all pixels are presented in the figures, this needs to be mentioned in the figures, e.g., by including a scale.
Lines 131-133: What was the name of the pulse sequence?
Line 148: What is 'NaN'? Some kind of chemical?
Lines 153-157: What was the name of the pulse sequence? Was the hole index calculated based on one MRI image only?
Lines 240-242: 'Consequently, CLM matured for 180 d revealed widely spaced and smaller eyes than CL matrices, which revealed holes that maintained or increased size throughout the RT.': Table 2 does not support this conclusion, more averages needed. At the very least, strong conclusions need to be avoided.
Table 3: Probably, these represent average parameters. Please mention this in the table.
It is generally easy to read, but may benefit from some careful editing.
Some specific issues:
Line 21: Delete 'Nevertheless,'
Line 204: Delete 'Regarding'
Author Response
Please, see attachment.

Reviewer 3 Report
Comments and Suggestions for Authors
The authors used magnetic resonance imaging (MRI) to study the structural changes and the evolution of textural characteristics of four types of Spanish feta cheese during ripening, both longitudinal and transverse. The data of the manuiscript is detailed and clear, and the overall structure is good, but there are several problems as follows:
1. Materials and methods: The description of the four kinds of cheese used is not particularly clear, whether the source is consistent;
2. For Figure1 and Figure2, the author mentioned in the manuscript that:As in the T2 maps, the structural anisotropy of the cheese matrix was also observed in the T1 maps (Fig. 2).
However, after comparison between the two figures, it is found that there is only a difference in brightness between them, and other texture profile are identical. I am a little skeptical about this result, please check it
Comments on the Quality of English LanguageEnglish expression is good.
Round 2
Reviewer 1 Report
Comments and Suggestions for Authors
The authors have not addressed the key criticisms I raised. The revised version remains largely unchanged regarding the NMR relaxometry data analysis and the citation of relevant literature. Significant flaws persist in the data analysis. I am confident that a proper analysis of the NMR relaxation data could yield different results. Consequently, I cannot consider the paper to be suitable for publication.
Reviewer 2 Report
Comments and Suggestions for Authors
The manuscript improved after revision.
Although, the authors addressed the question of this reviewer about the number of slices that were measured in their coverletter ("As described, 60 slices were considered (3 cm thickness each sample × 1 mm thickness
each slice × 2 samples/cheese"), the authors have not yet incorporated this information in the manuscript itself. The number of slices considered cannot be deduced from the information provided: 3 cm thickness each sample × 1 mm thickness each slice × 2 samples/cheese) does not imply 60 slices, as this would assume that there is no spacing between the slices and this is not necessarily always the case unless the acquisition parameters are tuned to that effect. Thus this should be clarified in the manuscript.
